# Impact of Environmental Regulation on Carbon Emissions in Countries along the Belt and Road—An Empirical Study Based on PSTR Model

**DOI:** 10.3390/ijerph20032164

**Published:** 2023-01-25

**Authors:** Lei Wu, Chengao Zhu, Xinhao Song, Junge He

**Affiliations:** 1School of Economics and Management, China University of Geosciences, Wuhan 430074, China; 2School of Physical Education, China University of Geosciences, Wuhan 430074, China; 3School of Foreign Languages, China University of Geosciences, Wuhan 430074, China; 4Center for Turkmenistan Studies, China University of Geosciences, Wuhan 430074, China

**Keywords:** environmental regulation, carbon emissions, PSTR, technical innovation, industrial structure, energy intensity

## Abstract

Since China has put forward the Belt and Road Initiative in 2013, research on the BRI-related countries along the Belt and Road has sprung up. With the advent of the era of carbon peak and carbon neutralization, environmental regulation, as one of the important methods to control carbon emissions, is becoming increasingly prominent. Research on the impact pathway of environmental regulation of countries along the Belt and Road on carbon emissions has important implications for environmental protection and carbon emission reduction. Based on the panel data of 38 countries along the Belt and Road from 2005 to 2018, this research applied linear Tobit model and nonlinear dynamic panel regression model (PSTR) to evaluate the direct impacts on carbon emissions from environmental regulation of countries along the Belt and Road, and to analyze the different impacts of environmental regulation on carbon emissions in terms of technical innovation, industrial structure, and energy intensity. We found that (1) the direct impact of environmental regulation on carbon emissions in the countries along the Belt and Road is positive, with slight differences between the Silk Road Economic Belt and 21st Century Maritime Silk Road (2) when technical innovation is at a low level, environmental regulation promotes carbon emissions, while at a high level, environmental regulation significantly inhibits carbon emissions. (3) When industrial structure is at both a low and high level, environmental regulation inhibits carbon emissions, with a stronger degree of inhibition at a higher level. (4) When energy intensity is at a low level, environmental regulation promotes carbon emissions, while at a high level, environmental regulation inhibits carbon emissions. Accordingly, we suggest that countries along the Belt and Road follow the road of sustainable and low-carbon development, which should further enhance their focus on environment protection, improve their environmental awareness, and take environmental regulation measures rationally to reduce carbon emissions. Meanwhile, relevant adjustments should be done on technical innovation, industrial structure, and energy intensity to achieve carbon emission reduction.

## 1. Instruction

Enlightened by the historical values of the ancient Silk Road, China has proposed the Belt and Road Initiative in 2013, aiming to develop an economic cooperation partnership with countries along the Belt and Road and promote the sustainable development of the global economy. Since then, with the economic development in BRI-related regions, serious resource and environmental problems are getting worse. At present, among the world’s leading carbon emitters, most of them are located along the Belt and Road. Carbon emissions in BRI-related countries have become a big challenge for the current global environmental and climate governance. Hence, environmental issues of countries along the Road have become a hot topic for research, and scholars spare no effort to conduct economic- and environment-related research on the countries along the Belt and Road. In addition, most countries along the Road are developing countries of irrational industrial structure, and their development is highly reliant on energy consumption, which causes massive emissions of CO_2_, seriously hindering further economic development. To realize sustainable and green economic growth, pollution issues must be handled first, and the most effective way to address environmental problems is to apply appropriate environmental regulation measures. However, since the environmental regulation effects are influenced by multiple factors, strengthening the degree of environmental regulation in an oversimplified way is not effective, which may bring unbearable and complex consequences, according to previous cases. Therefore, this research explored the possibility of tackling pollutions by environmental regulation measures and the effects of environmental regulation on carbon emissions under different intensities of influencing factors. This has a vital significance for dealing with the problems of economic development and environmental protection in countries along the Belt and Road, and promoting the achievement of global carbon emission reduction goals.

## 2. Literature Review

Environmental regulation, as an essential component of governmental regulation, is one of the vital measures to realize the coordination of environmental protection and economic development. Zhao (2009) considers environmental regulation as a binding force that aims to protect the environment for individuals or institutions in an existing form of visible systems or invisible consciousness [1]. Shen (2018) has classified the measures of environmental regulation into three categories: command-control environmental regulation, market-driven environmental regulation, and voluntary environmental regulation [2]. For command-control environmental regulation, the regulation measures include the investment in environmental pollution control, the licensing system of pollution discharge, and the environmental administrative punishment. For market-driven environmental regulation, the regulation methods include pollution fees, environmental protection tax, and eco-compensation. For voluntary environmental regulation, the regulation methods include environmental information disclosure, environmental labeling, public environmental supervision, environmental awareness education, and environmental complaint. In the meantime, carbon emission is a crucial conundrum all countries are facing. The exploration of environmental regulation approaches to handle carbon emissions has long been a hot topic for scholars to study. However, the current research results on the potential relationship between environmental regulation and carbon emissions are inconsistent.

On the one hand, the “Porter Hypothesis” recognized by many scholars shows that the application of environmental regulation can greatly improve the innovation level and offset governance cost of enterprises, and lead enterprises’ development to a high-tech, low-emission, and low-pollution pathway, effectively reducing carbon emissions. Wang and Liang (2022) have built an impulse response model of carbon emission patterns of micro enterprises to the intensity of environmental regulation. This model, using Chinese provincial-level panel data, points out that environmental regulation reduces China’s carbon emissions [3]. The lower the level of emission reduction technology, the higher the input ratio of emission reduction, and the higher the per capita income, the less significant the effects of environmental regulation on CO_2_ emission reduction. Khan (2019) examined the long-term and short-term relations of environmental regulation to carbon emissions using the ARDL model, and found that environmental regulation and carbon emissions have negative correlations in both the short- and long-terms. Therefore, the enhancement of green credit awareness, the improvement of technical innovation by R&D, and the rational use of current environmental regulation measures will help to realize the carbon emission reduction goal by 2030 [4]. Xu and Yang (2015) introduced the LMDI model and VAR model to analyze the effects of environmental regulation on China’s carbon emission reduction. Their results show that in the short-term, environmental regulation has inhibiting effects on China’s carbon emission increment and economic aggregate effect [5].

Nevertheless, some scholars point out that the application of environmental regulation policies aiming to reduce carbon emissions can potentially increase carbon emissions, namely, the Green Paradox hypothesis. This hypothesis has attracted broad attention due to its challenge to the effectiveness of environmental regulations. Currently, some studies in China show that environmental regulation and carbon emissions are nonlinearly related as an inverted U-shape. This indicates that environmental regulation, before its intensity reaches a certain level, will promote carbon emissions, while after its intensity reaches the certain level, it will significantly inhibit carbon emissions. Therefore, the relationship between environmental regulation and carbon emissions goes from Green Paradox effects to Forced Emission Reduction effects, which is being widely recognized by the academic circle. Yin (2022) took panel data from 30 Chinese provinces as samples to explore the indirect and direct effects of environmental regulation on carbon emissions, explaining the indirect effects through four conduction pathways in terms of energy consumption structure, industrial structure, technical innovation, and foreign direct investment (FDI) [6]. The empirical study suggests that the effects of environmental regulation on carbon emissions shows an inverted U-shaped curve, indicating that Green Paradox effects will be dominant when environmental regulation intensity is at a lower level, while with the tightening of environmental regulation policies, it will turn into Forced Emission Reduction effects on carbon emissions. Zhang and Wei (2014), using Chinese provincial-level panel data from 2000 to 2011, conducted an empirical analysis on the dual effects of environmental regulation on carbon emissions through a two-step GMM method [7]. The empirical study shows that environmental regulation affects carbon emissions with an inverted U-shaped curve. With the increase in environmental regulation intensity from weak to strong, the effects of environmental regulation on carbon emissions turn from Green Paradox to Forced Emission Reduction effects, and considering the actual condition of China, current environmental regulation policies effectively inhibit carbon emissions to reach the expected goal.

Meanwhile, some scholars go deeper into the indirect effects on carbon emissions from environmental regulation. Mei and Xin (2019) used the panel data of 30 Chinese provincial regions from 2000 to 2015 to build a panel threshold model with technical innovation as the threshold variable [8]. The study shows that technical innovation will influence the carbon emission reduction effects under environmental regulation. When technical innovation is at a higher level, environmental regulation will strengthen the effects on carbon emission reduction. Therefore, before developing environmental policies, technical innovation and other factors should be considered to design more targeted environmental regulation measures. Baloch (2022) adopted the latest data assessment technique of the common correlated effects group mean (CCEMG) to explore the effects of environmental regulation and financial development on carbon emissions of BRICS from 1995 to 2016, demonstrating that financial development promotes carbon emissions, and environmental regulations also have the same effects through stimulating carbon emissions, providing important insights for police makers to address environmental challenges [9]. Wang and Zuo (2018), using the panel data of China’s 30 provinces from 2004 to 2014, adopted the panel threshold model to analyze the threshold effects of China’s environmental regulation on carbon emissions and the regional differences in eastern, central, and western regions of China, with energy intensity, industrial structure, and FDI, respectively, as threshold variables [10]. The results indicate that, due to the role of different threshold variables, the impact of environmental regulation on carbon emissions shows a non-linear feature, and the policy effects of environmental regulation are significantly different in different regions. Wang and Wei (2020) used the panel data of 34 main members of OECD as samples to investigate the effects of environmental regulation on carbon emissions with technical innovation and environmental regulation as threshold values, respectively [11]. The result reveals that when the member of OECD has a higher technical innovation level, rebound effects will occur to increase CO_2_ emissions, and a higher environmental regulation level of emerging economies will cause serious Green Paradox effects and damage economic development. Li and Li (2021) chose the panel data from 2004 to 2019 to examine the threshold effect and spatial spillover effect of dual environmental regulation on carbon emission intensity under the limitation of technical innovation [12]. The result shows that at a middle or low technical innovation level, the Green Paradox effect of environmental regulation plays a dominant role, while at a high technical innovation level, Forced Emission Reduction effects become prominent and gradually offset the negative influences of Green Paradox effects under environmental regulation. The effects of formal environmental regulation on carbon emissions shows an inverted U-shaped double threshold feature.

In conclusion, whether the effects of environmental regulation on carbon emissions are linear or nonlinear, and whether the presentation of these effects are Green Paradox or Forced Emission Reduction, it should be illustrated based on the actual situation of the local region, because the relationship between environmental regulation and carbon emissions are not simply one-to-one interactions. At the same time, to study the effects of environmental regulation on carbon emissions, the specific application of spatial analysis or nonlinear analysis should be further discussed. Considering the effects of environmental regulation on carbon emissions, the view that Forced Emission Reduction effects follow the Green Paradox effects has been widely recognized by most scholars. That is, environmental regulation firstly promotes carbon emissions, and then inhibits carbon emissions with the increasing intensity of environment regulation. Whereas, the effects of environmental regulation on carbon emissions cannot be simply summarized into promotion first and inhibition next, in reality. The diversified empirical results embody different situations of the reality. In particular, countries along the Belt and Road chosen in this study, have different national conditions from that of China. Although there have been many related studies conducted previously, the studies about the effects of environmental regulation in countries along the Belt and Road and its indirect effects on carbon emissions have not been further presented. Therefore, the linear Tobit model and nonlinear PSTR model were adopted in this research to explore the linear and nonlinear effects of environmental regulation on carbon emissions in countries along the Belt and Road, and to explore the different effects of environmental regulation on carbon emissions with different threshold variables. Based on this, the paper classifies countries along the Belt and Road into countries, respectively, along the Silk Road Economic Belt and the Maritime Silk Road to discuss the heterogeneity of the effects of environmental regulation on carbon emissions of countries, respectively, along the two Silk Roads, and try to provide rational policy suggestions according to different national conditions.

## 3. Empirical Design

### 3.1. Model Building

#### 3.1.1. Tobit Model Building

Before the baseline model construction of the effects of environmental regulation on carbon emissions, influencing factors on carbon emissions are primarily studied. Nobuko (2004) used the input–output table to analyze the influencing factors of industrial CO_2_ emissions in Japan from 1985 to 1995, and took the influences of environment technical change (ETC) and production technical change (PTC) on carbon emissions into consideration [13]. Xu (2006) examined the influencing factors on Chinese carbon emissions, and classified the influencing factors of China’s per capita carbon emissions into energy structure, energy efficiency, economic development, and other aspects [14]. He (2012) adopted dynamic panel data to empirically study the influencing factors of industrial carbon emissions, and drew a conclusion that carbon emission reduction governance needs to have comprehensive considerations in terms of industrial structure, technical progress, macro-economic environment, and environmental regulation [15]. Fan (2019) calculated carbon emissions in Beijing Tongzhou District in China from 2008 to 2015 [16]. The study adopts the logarithmic mean exponential decomposition method to decompose the carbon emissions of the Tongzhou District into six influencing factors, which are population, per capita GDP, industrial structure, energy intensity, energy consumption structure, and other energy-related CO_2_ emission factors. According to the studies described above, it can be found that the influencing factors on carbon emissions center around population, economy, industrial structure, and energy intensity. Therefore, based on previous research, this study constructs the baseline model of environmental regulation on carbon emissions.

Meanwhile, due to the critical lack of carbon emission data of some countries in certain years, the empirical study uses data simulation to process the data. The inconsistent sources of carbon emission caliber, a large amount of compression of some observation data into one point, and the existence of some extreme values all seriously disturb the research results. In this case, the direct application of OLS regression (ordinary least squares regression) to evaluate the whole samples will cause errors and deviations of the results. Therefore, the panel Tobit model was adopted to perform basic regression analysis to eliminate the interferences of errors.

Considering the direct influences on carbon emissions from environmental regulation along the Belt and Road, the following function relationship was built:(1)$${\mathrm{CO}}_{2}=f\left(M\right)$$

CO_2_ refers to the carbon emissions of countries along the Belt and Road, and M refers to the influencing factors on carbon emissions of countries along the Belt and Road.

Due to the uncertainty of the function mentioned above, the relationship between carbon emissions and environmental regulation and other influencing factors cannot be observed directly, and a more accurate calculation model needs to be constructed. Considering the relations among variables, the specific Tobit model is built as follows:(2)$${\mathrm{CO}}_{2\mathrm{it}}=\beta_{0}+\beta_{i}{\mathrm{ER}}_{\mathrm{it}}+\beta_{r}X+u_{\mathrm{it}}$$

In this function, i refers to the country; t refers to time; CO_2it_ refers to the carbon emissions, an explanatory variable; ER_it_ refers to the level of environmental regulation; X refers to control variables, including development level, trading conditions, technical innovation, industrial structure, population, energy intensity, and other factors; and u_it_ refers to random errors.

#### 3.1.2. PSTR Model Building

After the examination of the linear relationship between environmental regulation and carbon emissions, the existence of the nonlinear relationship and the influencing mechanism between them was further explored. The PSTR model suggested by González (2017) was adopted to explore the nonlinear effects of environmental regulation on carbon emissions under different mechanisms. The PSTR model has following advantages [17]: (1) Compared to other nonlinear models, the PSTR model can reflect the transition speed of different mechanisms and better show the corresponding relationship among different variables by artificial introduction of the transfer function. (2) The PSTR model can be applied in a wider range to better reflect the data features of variables, regardless of the multicollinearity caused by introducing cross-terms, which has high applicability. Compared with other models, PSTR model can directly use the linearity test, and no differential heterogeneity test, to confirm the numbers of transfer functions and threshold variables in the model and further explore the interactions of economic variables. Different threshold variables were chosen to, respectively, study the influencing relationship between environmental regulation and carbon emissions. The relationship between them is examined and illustrated. The nonlinear PSTR model suitable for the research conditions is designed as follows:(3)$$Y_{\mathrm{it}}=\varphi_{i}+\delta_{1}X_{0\mathrm{it}}+\delta_{2}X+\overset{i}{\underset{J=1}{\sum }}\left(\beta_{1}X_{0\mathrm{it}}+\beta_{2}X\right)\times g_{i}\left(q_{\mathrm{it}};\gamma_{j},c_{\mathrm{jk}}\right)+u_{\mathrm{it}}$$

In this function, i and t, respectively, refer to the country and the particular year; $\varphi_{i}$ refers to increment, the individual fixation effect; $u_{\mathrm{it}}$ refers to the error term; $Y_{\mathrm{it}}$ refers to the explaining variable; $X_{0\mathrm{it}}$ refers to the explained variable, representing the data of the i province in the t year; X refers to the control variable; and q_it_ refers to the threshold variable.

The key part of this PSTR model is the transfer function ($q_{\mathrm{it}};\gamma_{j},c_{\mathrm{jk}})$, which is the continuously bounded function of the threshold variable q_it_. The range of the function is from 0 to 1, and the coefficient of the explained variable transfers between $\delta_{1}$ and $\delta_{1}$+$\beta_{1}$. According to Gonzalez’s design, the transfer function $g_{i}\left(\cdot \right)$ can be expressed as follows:(4)$$g_{i}\left(Y_{\mathrm{it}};\gamma_{j},c_{\mathrm{jk}}\right)={\left\{1+\exp\left[-\gamma_{j}\prod_{k=1}^{m}\left(q_{\mathrm{it}}-c_{\mathrm{jk}}\right)\right]\right\}}^{-1}$$

In this function, q_it_ refers to the threshold variable. Generally, for different threshold values, there are different relationships between the independent variable and the dependent variable. $\gamma_{j}$ refers to the coefficient of the j transfer function, which decides the transition speed of the transfer function from one state to another state. c refers to the location parameter. When the value of the threshold variable q_it_ equals that of the location parameter, the function will reach the fastest transition speed. The number of transfer functions r normally equals to 1 or 2, which classifies functions into low and high regime or low, middle, and high regime. At this point, the value of the transfer variable is greater or less than that of the location parameter. m refers to the number of location parameters, which is equal to 1 or 2. When m = 2, the minimum value of the transfer function will be from (c_1_ + c_2_)/2, and m equals to 1 when q_it_ obtains the maximum or minimum value. Particularly, when $\gamma_{j}\to \infty$, the transfer function $g_{i}\left(\cdot \right)$ will work as an exponential function, and the PSTR model will become a three-body PTR model. When $\gamma_{j}\to 0,$ the PSTR model will change into a fixed-effect panel model.

From the perspective of the study on the effects of environmental regulation on carbon emissions, the study, respectively, regards technical innovation, industrial structure, and energy intensity as three threshold variables, and designs the model as follows:(5)$${\mathrm{CO}}_{2\mathrm{it}}=\varphi_{i}+\delta_{1}{\mathrm{ER}}_{\mathrm{it}}+\delta_{2}X+\overset{i}{\underset{J=1}{\sum }}\left(\beta_{1i}{\mathrm{CO}}_{2\mathrm{it}}+\beta_{2i}X\right)\times g_{i}\left(q_{\mathrm{it}};\gamma_{j},c_{\mathrm{jk}}\right)+u_{\mathrm{it}}$$

In this function, the threshold variable $q_{\mathrm{it}}$ includes ${\mathrm{IV}}_{\mathrm{it}\;}$, ${\mathrm{IND}}_{\mathrm{it}}$, and ${\mathrm{EN}}_{\mathrm{it}}$. Other variables keep in consistence with the Tobit model. δ and β, respectively, refer to the coefficients in the linear part and nonlinear part.

Before testing the data in the PSTR model, the nonlinear relationship between environmental regulation and carbon emissions firstly needs to be examined, and statistical quantification of LM, ${\mathrm{LM}}_{F}$, and LRT will be constructed to test the linearity of the model.
(6)$$\mathrm{LM}=\frac{\mathrm{TN}\left({\mathrm{SSR}}_{0}-{\mathrm{SSR}}_{1}\right)}{{\mathrm{SSR}}_{0}}$$
(7)$${\mathrm{LM}}_{F}=\frac{\left({\mathrm{SSR}}_{0}-{\mathrm{SSR}}_{1}\right)/\mathrm{mk}}{{\mathrm{SSR}}_{0}/\left(\mathrm{TN}-N-\mathrm{mk}\right)}$$
(8)$$\mathrm{LRT}=-2\left[\log\left({\mathrm{SSR}}_{1}\right)-\log\left({\mathrm{SSR}}_{0}\right)\right]$$

Meanwhile, the model also needs to be tested for no surplus heterogeneity. The number of location parameters m can be decided according to the Akaike information criterion (AIC) and Bayesian information criterion (BIC). When r and m are fixed, the next evaluation is carried out with the PSTR model.

### 3.2. Variable Selection

The variables of this empirical model can be classified into four types: explained variable, explaining variable, control variable, and threshold variable.

Explained variable: The CO_2_ emissions (CO_2_). Per capita CO_2_ emissions are used to represent the CO_2_ emissions, namely per capita CO_2_ emissions = the total CO_2_ emissions/the total population. Due to the great differences among countries along the Belt and Road in terms of total population, economic development, and geographical conditions, this paper adopts per capita CO_2_ emissions as a substitute indicator rather than the total CO_2_ emissions.

Explaining variable: Environmental regulation (ER). The number of environmental technology invention patents was used to measure the level of environmental regulation. At present, although there are various methods to measure environmental regulation in environmental economics, the results of different methods are far from each other. Generally, three quantitative models were adopted for environmental regulation in previous research. The first measure is based on a qualitative index and traditionally scored by experts. The second is to measure directly by a quantitative index, such as pollution tax, expenditure on pollutant control, environmental tax, and environmental technology invention patents, etc. The final one is mainly to use entropy evaluation and factor analysis methods to construct a comprehensive quantitative index, such as the index combining input and output performance, and the comprehensive index obtained by standardizing weights of the solid, liquid, and gas wastes and various indexes. According to the actual situation of countries along the Belt and Road, a comprehensive quantitative index was adopted. However, due to the limitation of the data source, some indexes of countries along the Belt and Road are missing. Referring to the studies of Sarfraz et al. (2021), the study chose the number of environmental technology invention patents to represent the level of environmental regulation, which can effectively indicate the current situation of environmental regulation in different countries, since environmental technology invention patents always show the degree of emphasis on environmental protection in countries [18]. A larger number of patents shows more emphasis on environmental protection and also indicates a higher level of environmental regulation. This index has greater universality, especially applicable for countries along the Belt and Road with huge differences in national conditions.

Control variable: (1) Development level (DL); the per capita GDP of the host country (in constant 2010 prices) is used to indicate the local development level. Generally, per capita GDP can objectively reflect the level and degree of a country’s socio-economic development. Carbon emissions are closely related to humans’ economic activities. Scholars, such as Zhang (2017), found that the economic growth effect is the primary driving force for the increase in carbon emissions [19]. (2) Terms of trade (TR); the net barter terms of trade (NBTT) index is used to measure the foreign trade situation of the host country, which can be expressed as terms of trade index = export price index/import price index × 100. Considering that developing countries are dominant among countries along the Belt and Road, the terms of trade index is chosen to measure trade conditions of these countries, and to explore their trade conditions. In addition, due to the leading role of raw materials and mineral exports, the balance of trade has a weaker representation for reference in countries along the Belt and Road. Thus, this index is not adopted. (3) Technical innovation (IV); technical innovation generally refers to the innovation of production technology, including the development of new technology or the innovation of existing technologies in application. Most published papers measure technical innovation by the numbers of invention patents, while this paper decides to use the global invention index (GII) as the measure index of technical innovation on account of a lack of patents data of some countries along the Belt and Road. GII can measure the technical innovation level of the local area and reflect the local innovation level and economic development situation, which can also significantly impact carbon emissions. (4) Industrial structure (IND); industrial structure generally refers to the shares of agriculture, industry, and services in a country’s economic structure. Based on the situation of economic development and industrialization of countries along the Belt and Road, the proportion in GDP of the secondary industry is used to measure industrial structure. The secondary industry accounts for a larger proportion in all countries along the Belt and Road and better represents the local situation of their industrial structures. In general, heavy-polluting enterprises mainly concentrate in the secondary industry and have greater influences on carbon emissions. Talukdar found that an increased proportion of industries obviously promotes carbon emissions [20]. (5) Population (PO); the growth rate of the population of the host country is used as an index to measure demographic situation, reflecting the status and trend of population growth. The relationship is expressed as the growth rate of the population = (year-end population—year-begin population)/average population of the year × 1000%. Generally, the growth rate of the population can directly measure the increase in population and represent the health of the population structure of a country. Many studies have shown that population structure is closely related to carbon emissions. Liu (2022) analyzed the different influences on carbon emissions from various dimensions of population structure [21]. (6) Energy intensity (EN); the per capita kilogram of oil consumption is chosen as the index to measure energy intensity, because countries along the Belt and Road highly depend on oil and other fossil fuels. In general, the greater the energy intensity, the greater the per capita carbon emission intensity. In contrast, the smaller the energy intensity, the smaller the per capita carbon emission intensity. Due to the high dependance on oil and other fossil fuels of countries along the Belt and Road, per capita kilogram of oil consumption can better reflect the local energy consumption and carbon emission intensity.

Threshold variable: technical innovation (IV), industrial structure (IND), and energy intensity (EN).

In view of the study on the effects of environmental regulation on carbon emissions, and considering the main influencing factors of carbon emissions in countries along the Belt and Road, three threshold variables: technical innovation (IV), industrial structure (IND), and energy intensity (EN) are chosen in this study. From the perspective of the main influencing factors of carbon emissions, technical innovation has a direct impact on carbon emissions. IV is chosen as the threshold variable to further explore the role of the technical innovation level in the effects of environmental regulation on carbon emissions. In general, we expect that at a lower level of technical innovation, environmental regulation will promote carbon emissions, while at a higher level of technical innovation, environmental regulation will inhibit carbon emissions. As one of the key indices of measuring socio-economic situations of countries along the Belt and Road, industrial structure plays an important role in environmental governance. We expect that, at a lower level of industrial structure, environmental regulation will promote carbon emissions, while at a higher level of industrial structure, environmental regulation will inhibit carbon emissions. Energy intensity is a serious matter for countries along the Belt and Road, particularly some countries which are the origins of petroleum or have serious energy shortage problems. As a key index to measure energy situation, this paper expects that when energy intensity is at a lower level, environmental regulation will promote carbon emissions, while at a higher level, environmental regulation will inhibit carbon emissions.

### 3.3. Data Sources

There are 65 countries along the Belt and Road. To make the study more practicable, the research chooses the data of 38 countries along the Belt and Road, and further divides them into 11 countries along the Maritime Silk Road and 28 countries in the Silk Road Economic Belt, as Table 1 shows.

The research objectives of this paper, such as the index data of carbon emissions, development level, industrial structure, demographic situation, and energy intensity, are mainly from the World Bank Database (WB). The index data of environmental regulation are from the Organization for Economic Cooperation and Development database (OECD). The index data of terms of trade are from the Statistical Manual and data papers of the United Nations Conference on Trade and Development and International Financial Statistics of the International Monetary Fund (IMF). The technical innovation index is from the World Intellectual Property Organization Database (WIPO). Statistical descriptions of various variables are shown in Table 2.

Meanwhile, Figure 1 shows the situation of carbon emissions and environmental regulation of countries along the Belt and Road from 2005 to 2018, calculated by this research. According to the data of carbon emissions shown in Figure 1, carbon emissions of countries along the Silk Road Economic Belt is significantly higher than that of countries along the Maritime Silk Road. From 2008 to 2009, the carbon emissions of countries along the Silk Road have a drastic change, which sharply declined, then rapidly rebounded, declined again, and finally kept stable. Differently, the carbon emissions of countries along the Maritime Silk Road maintained a slow rising trend. From the perspective of the environmental regulation level, the overall trend of countries along the Belt and Road is rising first and then declining, including countries along the Maritime Silk Road, while the environmental regulation level of countries along the Silk Road Economic Belt is in a state of flux without a significantly changing trend.

## 4. Result Analysis

### 4.1. Tobit Results Analysis

The Tobit model was applied in this study to calculate the sample data of 38 countries along the Belt and Road via the software stata17.0, as Table 3 shows.

The regression results of the Tobit model show that, in general, the environmental regulation level of countries along the Belt and Road is significant, at 1%, indicating that at this point, environmental regulation has direct and positive effects on carbon emissions. The unreasonable energy consumption structure of countries along the Belt and Road is the main reason for the increased carbon emissions. In addition, some BRI-related countries, mostly developing countries, pay less attention to environmental protection, having great deficiencies in the choice of policy tools and the implementation of environmental protection policies, which restrict environmental regulation policies for exerting a positive influence on carbon reduction. For rapid economic growth, they have to loosen the enforcement of environmental regulation, seeking “resource rent”.

The environmental regulation of countries along the Maritime Silk Road is positively significant at the 5% level, indicating that environmental regulation has positive promotional effects on carbon emissions. The elevated environmental regulation level will promote the increase in carbon emissions. However, the coefficient is significantly lower than 0.045 on the whole. The reason may be that countries along the Maritime Silk Road are more affected by foreign trade and foreign capital, and pay more attention to environmental protection issues, which can effectively reduce carbon emissions.

The environmental regulation coefficient of countries along the Land Silk Road is negative, and does not pass the significance test, showing that there are no significant effects on carbon emissions from environmental regulation. These countries may have awareness to reduce carbon emissions through environmental regulation, but environmental regulation measures are restricted for their economic benefits and national industrialization. Furthermore, the interaction of environmental regulation and carbon emissions are not the simple linear relationship, and inhibition only shows part of the effects on carbon emissions by environmental regulation, which leads to the insignificant result.

According to the test results above, the effects of environmental regulation on carbon emissions of countries along the Belt and Road are not fully revealed, which suggests that a simple linear relationship cannot fully demonstrate the effects of environmental regulation on carbon emissions. Therefore, to test the nonlinear effects of environmental regulations of countries along the Belt and Road, the PSTR model was used to further explore the effects of the two factors.

The study highlights the effect of environmental regulation on carbon emissions, and other control variables are not the key research objects. Their coefficients are checked to verity the consistency with the results gained from PSTR model.

### 4.2. PSTR Results Analysis

#### 4.2.1. Parameterization of PSTR Model

According to the results of the linear test in Table 4, ${\mathrm{IV}}_{\mathrm{it}}$, ${\mathrm{IND}}_{\mathrm{it}}$, and ${\mathrm{EN}}_{\mathrm{it}}$, as threshold variables, all reject the original hypothesis, demonstrating the existence of a nonlinear relationship. According to the residual nonlinear results of Table 5, when the threshold variables are ${\mathrm{IV}}_{\mathrm{it}}$ and ${\mathrm{EN}}_{\mathrm{it}}$, the best choice is r = 1, which means there is one transfer function. When ${\mathrm{IND}}_{\mathrm{it}}$ is chosen as the threshold variable, the nonlinear relationship is too complex because of the r greater than 2. Considering the principle of a smaller value chosen for ${\mathrm{LM}}_{F}$ and the optimal value chosen for PSTR model results, therefore, the best choice is r = 2, which means the existence of two transfer functions. In the defined table of model parameterization, based on ATC and BIC minimization criterion, it can be shown that when ${\mathrm{IND}}_{\mathrm{it}}$ and ${\mathrm{EN}}_{\mathrm{it}}$ choose m = 1, which means there is only one m, while ${\mathrm{IV}}_{\mathrm{it}}$ should choose m = 2, as Table 6 shows. Due to the location parameters and results both out of the rational limitations when m = 2, the research finally chooses m = 1 rather than m = 2. Therefore, when the threshold variable is ${\mathrm{IND}}_{\mathrm{it}}$, r = 2, m = 1; when the threshold variable is ${\mathrm{IV}}_{\mathrm{it}}$ or ${\mathrm{IV}}_{\mathrm{it}}$, r = 1, m = 1.

#### 4.2.2. PSTR Results Analysis with Technical Innovation as the Threshold Variable

As technical innovation is chosen as the threshold variable, the result has one transfer mechanism. When 0 < IV < 33.8705 in the low regime, the speed of the transfer function will increase as IV increases, while when I > 33.8705 in the high regime, the speed of the transfer function will decrease as IV increases. At the point of 33.8705, IV realizes the transition from the low regime to high regime with the highest slope of the transfer function. The slope coefficient $\gamma_{1}$ = 0.2254 refers to a smoother transfer regime of the model. in statistical descriptions of variables, the average value of IV is 34.985, while that of location parameter is $c_{1}$ = 33.8705, which shows that most observed values are in the high regime. However, due to the small difference, the distribution of the sample quantity is relatively uniform. The results are shown in Table 7.

From Figure 2, it can be found that the transfer function image of environmental regulation demonstrates an S shape. When IV is at a lower level, the value of the transfer function is close to 0, and the effects of environmental regulation on carbon emissions are mainly presented in the linear part. According to the Table above, when taking technical innovation as the threshold variable, the coefficient in the linear part is positive, while negative in the nonlinear part, and both of the two parts show significant results at a 5% level, illustrating that when technical innovation is at a lower level, environmental regulation will promote carbon emissions. Nonetheless, when technical innovation is at a higher level, the coefficient of effects of environmental regulation on carbon emissions will transfer from positive to negative, and environmental regulation will inhibit carbon emissions. The Green Paradox effect and Forced Emission Reduction effect may be responsible for the result, which is also consistent with the results of Zhang and Wei’s (2014) research [7]. At this point, the impact of environmental regulation on carbon emissions presents an inverted “U” shape.

When the technical innovation of countries along the Belt and Road is at a low level, the promulgation of environmental regulation policies cannot offset high carbon emissions brought by extensive economic growth with low technology. The effects of environmental regulation are not significant in inhibiting carbon emissions through technological innovation, with the Green Paradox effect playing a dominant role. The countries, enterprises, and other main players along the Belt and Road are unable to introduce high and new technology or use technological innovation to reduce their cost in carbon emission reduction, but instead increase their production cost, which forces these enterprises to put more emphasis on the extensive economic growth mode to increase benefits at the cost of resources and environment, and the carbon emissions will be further increased.

When technical innovation is at a high level, the effects of environmental regulation on carbon emissions will show a significant Forced Emission Reduction effect. In this period, under a high technical innovation level, government and enterprises can use high technology of energy saving and emission reduction to offset the negative effects produced by the Green Paradox, gradually. The effects of environmental regulation on carbon emissions are staying in the falling stage of the inverted U curve and will formally transfer into inhibiting effects on carbon emissions, which will be enhanced if the environmental regulation level is continuously increased.

#### 4.2.3. PSTR Results Analysis with Industrial Structure as the Threshold Variable

When industrial structure is chosen as the threshold variable, the results will have two transfer mechanisms, but also two basically equal location parameters, which means, in the middle regime, there are no efficient values as 26.3883 < IV < 26.6685, and middle regime values do not exist when industrial structure is used as the threshold variable. However, the negative and significant coefficients in both low and high regimes indicate that environmental regulation has inhibiting effects on the increase in carbon emissions regardless of the proportion of the secondary industry. With the gradual increase in the industrial structure level, the coefficient slowly changes from −0.0351 to −0.0899, and the coefficient in the high regime is less than that in the low regime, which shows the increasing effects of inhibition on carbon emissions. In statistical descriptions of variables, the average value of IND is 30.251 and the location parameter value is 26. This suggests that most observed values are distributed in the high regime, slightly different from the distribution of samples. The results are shown in Table 8.

As shown in Figure 3, the transfer function image of the industrial structure does not present a significant feature. When IND is at a higher level, the value of the transfer function is close to 1, and the effects of environmental regulation on carbon emissions is reflected from the sum of linear and nonlinear parts. From Table 8, it can be observed that the values of both linear and nonlinear parts are negative, which means generally negative effects on carbon emissions. The slope parameters of the two parts are both less than 1, revealing a smoother transition between model regimes.

When industrial structure in countries along the Belt and Road is at a low or high level, environmental regulation will have significant inhabiting effects on carbon emissions, which exhibits an inhibition on carbon emissions from environmental regulation at any kind of industrial structure level. According to the theory of industrial structure, economic development is a changing process of a series of interrelated economic structures. Generally, industrial structure adjustment is a transfer process of industrial structure from “primary industries, tertiary industries, secondary industries” type to “secondary industries, tertiary industries, primary industries” type, and finally “tertiary industries, secondary industries, primary industries” type. According to the opening P curve theory of environmental regulation on carbon emissions put forward by scholars including Chen (2013), Figure 4 shows the effects of environmental regulation on carbon emissions in countries along the Belt and Road when industrial structure is taken as the threshold variable [22]. As the countries along the Belt and Road chosen in this research are mainly developing countries, their industrial structure has not reached the advanced level and is on the upspring stage of the P curve.

When industrial structure is at a low level, the economic development of the host country is in the “primary industries, tertiary industries, secondary industries” type stage with a lower level of industrialization, and causes carbon emissions. However, domestic requirements for environmental protection and carbon emissions are low, and the corresponding environmental regulation policies are not strictly implemented. Thus, environmental regulation has played an inhibitory role in carbon emissions, but not strong enough, which is a deviation from the expectation in this study that under a lower industrial structure, environmental regulation will promote carbon emissions. The changes in industrial structure reflect the process of industrialization, which definitely causes the increase in carbon emissions. Accordingly, when industrial structure is at a high level, the economic development of the host country has been in the “secondary industries, tertiary industries, primary industries” type stage with an improved industrial structure level. The increase in environmental pollution and sharp rise in carbon emissions also aggravate the environment problems of the country. Therefore, more powerful policies of environmental regulation need to be introduced to strengthen the inhabiting effects on carbon emissions to meet domestic environmental protection requirements.

#### 4.2.4. PSTR Results Analysis with Energy Intensity as the Threshold Variable

When taking energy intensity as the threshold variable, the research result has only one transfer mechanism. When 0 < EN < 4.9365, it belongs to the low regime, while when EN > 4.9365, it is at the high regime. At the point of 4.9365, EN has realized the transition from the low regime to the high regime, and the largest slope of the transfer function is 5.9163, a smoother transfer. The average value of EN is 19.887, while the location parameter value is $c_{1}$ = 4.9365. This means that most values of EN are on the right side of the location parameters, and namely in the high regime. Due to the larger distribution deviation of samples, there is a significant difference between the low regime and high regime. The results are shown in Table 9.

As Figure 5 shows, the transfer function image of energy intensity presents a horizontal L shape. When EN is at a higher level, the whole value is close to 1 and the effects of environmental regulation on carbon emissions are reflected on the sum of the linear part and nonlinear part. According to Table 9, when EN is at the low regime, environmental regulation has positive effects on carbon emissions, and when EN is at the high regime, environmental regulation has negative effects on carbon emissions, and the two situations both show significance under 5%. Namely, when energy intensity is at a lower level, environmental regulation promotes carbon emissions, and when energy intensity is at a higher level, environmental regulation will inhibit carbon emissions. Meanwhile, the effects of environmental regulation on carbon emissions shows an inverted U shape, which is possibly because of the selection of environmental regulation to reduce pollution and decrease emissions, or the two-way selection effect of environmental regulation. If the observed slope parameter is $\gamma_{1}$ = 5.9163, the function can transfer smoothly. However, on account of the rare quantity of observed values, the image shows an unsmooth and irregular shape. At a low energy intensity level, environmental regulation will have positive effects on carbon emissions, which means environmental regulation promotes carbon emissions. Since the low consumption of energy leads to a lower level of carbon emissions, the introduction of environmental regulation policies at this stage possibly aims to control pollutant discharges rather than reduce carbon emissions. Enterprises and other pollutant bodies, due to extra costs of pollution control, are bound to expand scales of operation to offset corresponding financial loss and maximize their economic and social benefits, which definitely increases carbon emissions. The “two-way choice system” effect for pollution reduction leads to the increase in carbon emissions.

With the ongoing economic activities, energy consumption will reach a new high. In this condition, when energy intensity is at the high regime, a large amount of carbon dioxide and other pollutants will be produced by energy consumption and various economic activities, including huge negative external effects from pollutants pressing for emission reduction. At this stage, new environmental regulation policies will absolutely plan carbon emission reduction as one of their main goals. Therefore, environmental regulation will negatively inhibit carbon emissions, indicating that the “two-way choice system” effect for carbon reduction decreases carbon emissions.

## 5. Robustness Test

### 5.1. Substituting the Explained Variable

In order to test the robustness of the Tobit model results, based on the study of Zhang (2012), environmental tax was used to substitute environmental technical innovation patents as the substitute indicator of environmental regulation in this paper, to reoperate the test process of the above hypothesis [23]. However, the data of environmental tax are from the OECD Database, which only includes OECD countries along the Silk Road, while other countries of the research sample are not involved. Thus, due to the loss of the environmental tax data of some countries, the paper only chose to test the Tobit model results of some countries along the Land Silk Road and the Marine Silk Road to ensure the reasonability of the robustness test. The results are shown in Table 10. Compared with the results of Table 3, the regression results are basically consistent.

According to the test results, the influence coefficient of environmental regulation on carbon emissions in countries along the Land Silk Road is still negative, consistent with the results shown in Table 3. Its significance is also improved, passing the test at the 1% significance level. The influence coefficient of environmental regulation on carbon emissions in countries along the Marine Silk Road is still positive, passing the test at the 5% significance level, consistent with the results shown in Table 3. The above robustness test results indicates that the conclusion of this research has a better robustness.

### 5.2. Winsorize Test

Due to the existence of extreme values in variables, such as the environmental regulation index, to eliminate the influence of extreme factors on PSTR model, the study, taking the research method of scholars like Chen (2020) for reference, deals with the data by winsorize, and addresses all indexes by bilateral winsorize on 1% quartile to further examine the robustness of the conclusions mentioned above [24].The results are shown in Table 11. The comparison of the regression results of Table 7, Table 8 and Table 9, in turn, reveals that the regression results are about the same.

In terms of IV, the coefficient of environmental regulation on carbon emissions is positive when IV is in the low regime, and negative when IV is in the high regime, which are both significant at 5%. Namely, environmental regulation promotes carbon emissions in the low regime and inhibits carbon emissions in the high regime, which is consistent with the results involving control variables. On this ground, the result is robust. When the threshold variable is IND, there are also three regimes, similarly, with negative values in the low and high regimes showing significance at 5% and 10%, respectively, and negative values in the middle regime, which means environmental regulation on carbon emissions presents the inhibition–promotion–inhibition effect mechanism. The result is consistent with the result involving control values, which indicates its robustness. Although the middle regime does not pass the significance, the results are also corresponding to the research results above. Similarly, when the threshold variable is EN, the coefficient is positive in the low regime and negative in the high regime, which are both significant at 5%, consistent with the result above. This also demonstrates the robustness of the results.

## 6. Conclusions

The relationship between environmental regulation and carbon emissions has always been a research highlight in resource and environment economics. Based on the panel data of 38 countries along the Road from 2005 to 2018, and considering the regional heterogeneity of development in countries along the Belt and Road, this study applies the nonlinear dynamic panel regression model (RSTR) to explore the nonlinear relationship between environmental regulation and carbon emissions of countries along the Belt and Road from the three perspectives of technical innovation, industrial structure, and energy intensity. The following conclusions have been reached.

First, in the research samples, environmental regulation and carbon emissions have a significant nonlinear relationship. When technical innovation is used as the threshold variable, environmental regulation will promote carbon emissions in the low regime and inhibit carbon emissions in the high regime. When industrial structure is used as the threshold variable, environmental regulation will inhibit carbon emissions in both low and high regimes at different degrees, while there are no samples to describe the promoting effects in the middle regime. When energy intensity is used as the threshold variable, the effects of environmental regulation on carbon emissions are also positive in the low regime and negative in the high regime.

Second, different threshold variables lead to the promotion of carbon emissions from environmental regulation for different reasons. When the threshold variable is technical innovation, the main reasons for the changing effects on carbon emissions from environmental regulation are the Green Paradox effect and Forced Emissions Reduction effect. When the threshold variable is industrial structure, the main reason for the different degrees of inhibition of carbon emissions from environmental regulation is the economic cost from the development of the industrial structure. When the threshold variable is energy intensity, the main reason for the effects on carbon emissions from environmental regulation is the “two-way selection” effect. Under different threshold variables, the effects of environmental regulation on carbon emissions are driven by different factors.

Third, the study uses winsorize to conduct robustness test of the research, which achieves basically consistent results with the original model results, and the positive and negative test and significant test of each regime all essentially pass, proving that the different effects of environmental regulation on carbon emissions are robust under different threshold variables of technical innovation, industrial structure, and energy intensity, and providing a further supplement for the effect mechanism of environmental regulation on carbon emissions.

According to the conclusions above, this paper suggests (1) taking environmental regulation measures rationally. Firstly, countries along the Belt and Road must attach greater importance to environmental protection. They need to enhance the construction of environmental laws and regulations, strengthen public education on environment protection, build environmental quality supervision mechanisms, and stick to the concept of green and environment-friendly development. Secondly, environmental regulation should be implemented at a proper intensity. Both lower and higher levels of environmental regulation will be not conducive to address environment issues. Moderate environmental regulation will play an important role in effectively promoting sustainable and green development. Thirdly, various kinds of environmental regulation measures need to be improved. Control-oriented environmental regulation tools such as environment tax and pollution discharge limit should be combined with emission trading, environmental subsidies, and other motivational environmental regulation tools. Meanwhile, the carbon emission trading market and other economic measures need to be put to good use to fully leverage the effects of different environmental regulation tools on energy conservation and emission reduction. (2) Adopting differentiated environmental regulation measures according to the regional conditions. To handle environment problems in countries along the Belt and road, countries with a lower technical innovation level should not only vigorously introduce foreign advanced technology, but also promote domestic technical innovation through proper environmental regulation policies to improve the level of science and technology. Countries with unreasonable industrial structures need to adjust their industrial structures. In particular, the too-high or too-low proposition of the secondary industry should be avoided, which is not conducive to addressing environmental problems. Countries with excessively high energy intensity should modestly reduce the use of fossil energy and energetically develop clean energy to reduce carbon emissions and maximize the benefits of environmental protection. (3) Promoting high quality and comprehensive development by measures suitable for national conditions. Various influencing factors, such as national economic development, population, and trading conditions, all have effects on carbon emissions at different degrees. It is not realistic to deal with carbon emission issues only by environmental regulation measures. Therefore, BRI-related countries should deeply explore the main sources of carbon emissions, and combine environmental policies with industrial policies, foreign trade policies, and technical innovation promoting policies in a coordinated way to realize an all-round and high-quality development of the country’s social economy. Only in this way can the carbon emission issues of the country be dealt with fundamentally.

The Tobit and PSTR models were applied in this study to analyze the linear and nonlinear effects of environmental regulation on carbon emissions. However, a more complicated statistical model was restricted for the incompleteness and unavailability of some data in countries along the Belt and Road. In addition, further exploration into the classification of the effects of environmental regulation on carbon emissions of countries along the Belt and Road will be the focus of future research on the relationship between environmental regulation and carbon emissions.

## Figures and Tables

**Figure 1 ijerph-20-02164-f001:**
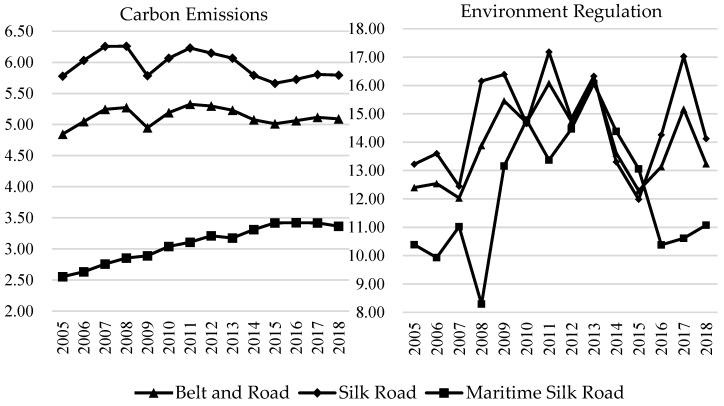
The situation of carbon emissions and environmental regulation of countries along the Belt and Road.

**Figure 2 ijerph-20-02164-f002:**
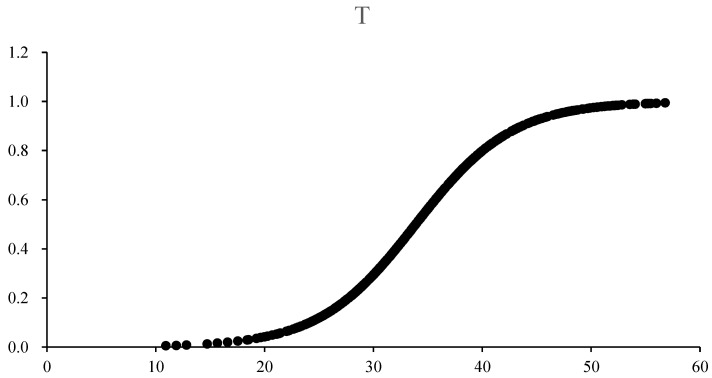
The transfer function image of technological innovation.

**Figure 3 ijerph-20-02164-f003:**
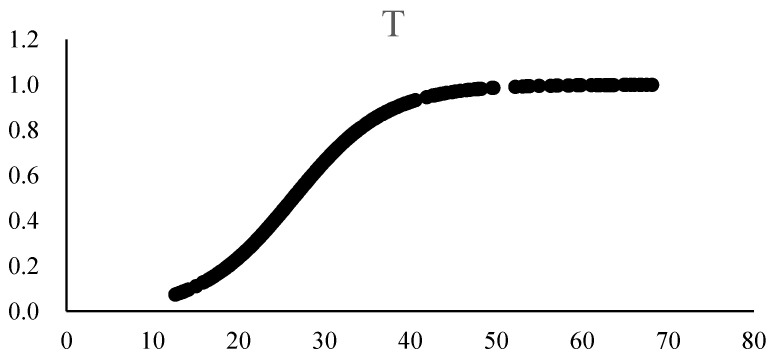
The transfer function image of industrial structure.

**Figure 4 ijerph-20-02164-f004:**
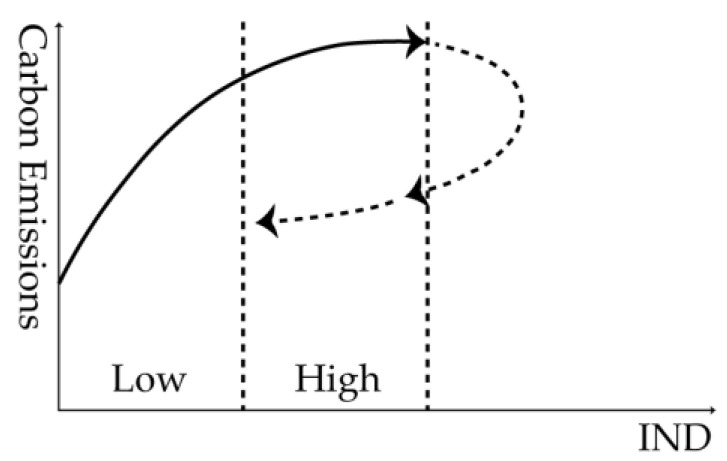
Opening P curve of environmental regulation on carbon emissions.

**Figure 5 ijerph-20-02164-f005:**
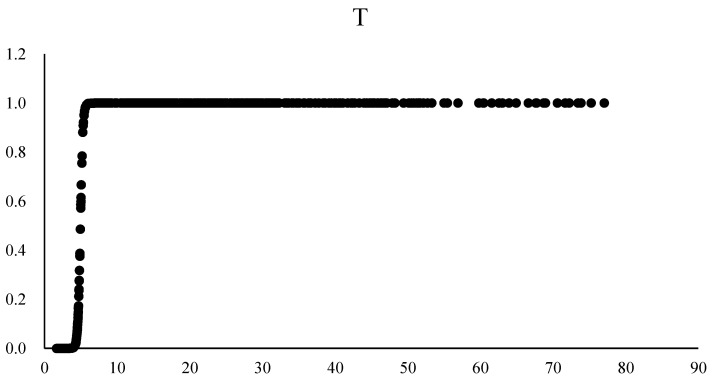
The transfer function image of energy intensity.

**Table 1 ijerph-20-02164-t001:** 38 countries along the Belt and Road covered in this paper.

Countries along the Land Silk Road	Countries along the Maritime Silk Road
Estonia, Croatia, Czech Republic, Hungary, Israel, Latvia, Oman, Russia, Slovakia, Slovenia, Albania, Azerbaijan, Belarus, Bosnia and Herzegovina, Bulgaria, Georgia, Iran, Jordan, Kazakhstan, Kyrgyzstan, Moldova, Mongolia, North Macedonia, Romania, Ukraine, Uzbekistan, Nepal	India, Indonesia, Saudi Arabia, Vietnam, Thailand, Bangladesh, Egypt, Pakistan, Turkey, Philippines, Sri Lanka

**Table 2 ijerph-20-02164-t002:** Statistical descriptions of variables.

Variable	Observed Value	Average Value	Standard Deviation	Maximum Value	Minimum Value
CO_2_ emissions (CO_2_)	532	5.126	4.107	17.692	0.101
Environmental Regulation (ER)	532	14.003	7.843	55.290	1.030
Development Level (DL)	532	8.069	7.729	42.063	0.316
Terms of Trade (TR)	532	112.779	35.540	254.017	46.276
Technical Innovation (IV)	532	34.985	8.308	56.790	10.925
Industrial Structure (IND)	532	30.251	10.777	68.187	12.655
Population (PO)	532	0.822	1.370	7.350	−2.081
Energy Intensity (EN)	532	19.887	16.187	77.040	1.639

**Table 3 ijerph-20-02164-t003:** Regression results of Tobit.

Variable	Along the Belt and Road	Along the Silk Road	Along the Maritime Silk Road
ER	0.045 *** (3.23)	−0.006 (−0.32)	0.027 ** (2.06)
DL	0.343 *** (23.95)	0.261 *** (14.69)	0.543 *** (26.85)
TR	0.029 *** (8.14)	0.029 *** (6.39)	−0.009 ** (−2.18)
IV	0.019 (1.23)	0.013 (0.72)	0.004 (0.32)
IND	0.084 *** (6.47)	0.086 *** (5.31)	0.109 *** (7.60)
PO	−0.130 (−1.48)	−0.064 (−0.63)	0.646 *** (4.42)
EN	0.009 (1.19)	0.012 (1.24)	−0.001 (−0.18)
_cons	−4.826 *** (−7.60)	−3.007 *** (−3.58)	−3.634 *** (−7.62)

Notes: (1) the figure in brackets refers to the Z statistics of the coefficients; (2) **, and ***, respectively, refer to the striking results under the significant level of 5%, and 1%.

**Table 4 ijerph-20-02164-t004:** Linear test.

Threshold Variable	$H_{0}$: r = 0, $H_{1}$: r = 1.		
	LM	${\mathrm{LM}}_{F}$	LRT
${\mathrm{IV}}_{\mathrm{it}}$	19.800 *** (0.006)	2.689 *** (0.010)	20.178 *** (0.005)
${\mathrm{IND}}_{\mathrm{it}}$	148.918 *** (0.000)	27.045 *** (0.000)	174.706 *** (0.000)
${\mathrm{EN}}_{\mathrm{it}}$	12.657 * (0.081)	1.696 (0.108)	12.810 * (0.077)

Notes: (1) the figure in brackets refers to the Z statistics of the coefficients; (2) *,and ***, respectively, refer to the striking results under the significant level of 10%, and 1%.

**Table 5 ijerph-20-02164-t005:** Residual nonlinear test.

Threshold Variable	$H_{0}$: r = 1, $H_{1}$: r = 2			$H_{0}$: r = 2, $H_{1}$: r = 3		
	LM	${\mathrm{LM}}_{F}$	LRT	LM	${\mathrm{LM}}_{F}$	LRT
${\mathrm{IV}}_{\mathrm{it}}$	6.532 (0.479)	0.840 (0.554)	6.573 (0.475)			
${\mathrm{IND}}_{\mathrm{it}}$	41.523 *** (0.000)	5.721 *** (0.000)	43.233 *** (0.000)	43.862 *** (0.000)	5.982 *** (0.000)	45.776 ** (0.000)
${\mathrm{EN}}_{\mathrm{it}}$	5.217 (0.634)	0.669 (0.698)	5.243 (0.630)			

Notes: (1) the figure in brackets refers to the Z statistics of the coefficients; (2) **, and ***, respectively, refer to the striking results under the significant level of 5%, and 1%.

**Table 6 ijerph-20-02164-t006:** Model parameterization.

Threshold Variable	${\mathrm{IV}}_{\mathrm{it}}$		${\mathrm{IND}}_{\mathrm{it}}$		${\mathrm{EN}}_{\mathrm{it}}$	
	m = 1	m = 2	m = 1	m = 2	m = 1	m = 2
AIC	1.794	1.785	1.265	1.545	1.742	1.748
BIC	1.922	1.921	1.466	1.681	1.871	1.884

**Table 7 ijerph-20-02164-t007:** PSTR results with technical innovation as the threshold variable.

Variable	${\mathrm{IV}}_{\mathrm{it}}$	
	Linear	Nonlinear
ER	0.0985 ** (3.1521)	−0.0930 ** (−2.0522)
GDP	0.2755 ** (6.8752)	0.1559 * (2.4228)
TR	0.0246 *** (4.0952)	0.0156 ** (1.3005)
IV	0.1421 * (1.5081)	−0.0365 * (−0.5755)
IND	0.1506 ** (4.8156)	−0.1237 ** (−2.7391)
PO	−0.1519 (−0.8885)	−0.0056 (−0.0208)
EN	0.0196 ** (0.7828)	−0.0128 ** (−0.2803)
γ	$\gamma_{1}$ = 0.2254	
c	$c_{1}$ = 33.8705	
RSS	2915.747	

Notes: (1) the figure in brackets refers to the Z statistics of the coefficients; (2) *, **, and ***, respectively, refer to the striking results under the significant level of 10%, 5%, and 1%.

**Table 8 ijerph-20-02164-t008:** PSTR results with industrial structure as the threshold variable.

Variable	${\mathrm{IND}}_{\mathrm{it}}$		
	Linear	Nonlinear	Nonlinear
ER	−0.0351 ** (−1.3136)	0.1689 * (1.6929)	−0.0899 * (−1.4114)
GDP	−0.2177 ** (−6.5934)	1.8037 (13.7181)	−0.6932 * (−9.3617)
TR	0.0409 *** (5.7369)	−0.1099 ** (−4.4354)	0.0993 ** (6.8646)
IV	−0.0415 ** (−1.4575)	0.1029 * (1.2529)	0.0037 * (0.0644)
IND	−0.0551 * (−0.7902)	0.1705 (1.6539)	−0.2863 * (−3.8409)
PO	−0.1951 (−0.8686)	−0.0977 (−0.1294)	−0.0658 (−0.1443)
EN	−0.0161 ** (−0.8710)	0.0867 * (1.5428)	−0.0403 ** (−1.1418)
γ	$\gamma_{1}$ = 0.1840/1.0423		
c	$c_{1}$ = 26.3883; $c_{2}$ = 26.6685		
RSS	1632.193		

Notes: (1) the figure in brackets refers to the Z statistics of the coefficients; (2) *, **, and ***, respectively, refer to the striking results under the significant level of 10%, 5%, and 1%.

**Table 9 ijerph-20-02164-t009:** PSTR results with energy intensity as the threshold variable.

Variable	${\mathrm{EN}}_{\mathrm{it}}$	
	Linear	Nonlinear
ER	0.1378 ** (3.4843)	−0.1083 ** (−2.5239)
GDP	0.2178 ** (6.6439)	0.1625 ** (3.8652)
TR	0.0190 *** (2.6139)	0.0193 *** (2.2686)
IV	0.0768 * (1.1043)	−0.0389 * (−0.5982)
IND	0.2032 ** (6.0706)	−0.1532 ** (−4.0773)
PO	−0.2397 (1.1653)	0.0762 (0.3408)
EN	−0.3319 (−0.7428)	0.3505 (0.7870)
γ	$\gamma_{1}$ = 5.9163	
c	$c_{1}$ = 4.9365	
RSS	2769.487	

Notes: (1) the figure in brackets refers to the Z statistics of the coefficients; (2) *, **, and ***, respectively, refer to the striking results under the significant level of 10%, 5%, and 1%.

**Table 10 ijerph-20-02164-t010:** Tobit regression results.

Variable	Along the Land Silk Road	Along the Marine Silk Road
ER	−0.180 *** (−3.11)	0.093 ** (2.17)
DL	0.059 ** (2.41)	0.145 *** (3.04)
TR	0.025 *** (5.65)	0.028 *** (4.86)
IV	0.018 *** (2.91)	−0.011 ** (−2.27)
IND	0.035 (1.62)	0.040 * (1.92)
PO	0.088 (0.60)	0.048 (0.22)
EN	0.009 *** (3.18)	−0.003 (−1.30)
_cons	1.186 (0.98)	−2.158 ** (−2.06)

Notes: (1) the figure in brackets refers to the Z statistics of the coefficients; (2) *, **, and ***, respectively, refer to the striking results under the significant level of 10%, 5%, and 1%.

**Table 11 ijerph-20-02164-t011:** Robustness test results.

Variable	${\mathrm{IV}}_{\mathrm{it}}$		${\mathrm{IND}}_{\mathrm{it}}$			${\mathrm{EN}}_{\mathrm{it}}$	
	Linear	Nonlinear	Linear	Nonlinear	Nonlinear	Linear	Nonlinear
ER	0.0893 ** (3.9672)	−0.0849 ** (−2.8130)	−0.0276 ** (−1.0961)	0.1693 (1.6875)	−0.0977 * (−1.4564)	0.1295 ** (3.4130)	−0.0939 ** (−2.2367)
GDP	0.3399 ** (13.3586)	0.0646 ** (1.6709)	−0.1478 ** (−4.9741)	1.6922 (13.5823)	−0.6772 * (−9.2867)	0.2379 ** (7.0604)	0.1507 ** (3.6441)
TR	0.0306 *** (7.8047)	0.0094 *** (1.1776)	0.0353 *** (5.3571)	−0.0949 ** (−3.9834)	0.0934 ** (6.4972)	0.0208 *** (2.9397)	0.0178 *** (2.1345)
IV	0.0262 ** (0.6682)	0.0196 ** (0.6670)	−0.0270 ** (−0.9841)	0.0724 * (0.9065)	0.0170 * (0.2943)	0.0503 * (0.6617)	−0.0159 * (−0.2260)
IND	0.1009 ** (4.8012)	−0.0582 ** (−2.0277)	−0.0278 * (−0.4252)	0.1271 (1.2365)	−0.2690 * (−3.4816)	0.1914 ** (6.0488)	−0.1420 ** (−3.9823)
PO	−0.0984 (−0.9005)	−0.1179 (−0.6983)	−0.2879 (−1.3177)	−0.2355 (0.3058)	−0.2066 (−0.4478)	−0.2050 (−0.9686)	−0.0505 (0.2194)
EN	0.0151 ** (1.0912)	0.0002 ** (0.0086)	−0.0165 ** (−0.9453)	0.0935 * (1.6933)	−0.0453 ** (−1.2608)	−0.1606 (−0.3406)	0.1786 (0.3799)

Notes: (1) the figure in brackets refers to the Z statistics of the coefficients; (2) *, **, and ***, respectively, refer to the striking results under the significant level of 10%, 5%, and 1%.

## Data Availability

Not applicable.

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
