# Peer review of "Impact of Environmental Regulation on Carbon Emissions in Countries along the Belt and Road—An Empirical Study Based on PSTR Model"

_ijerph, 2023, doi:10.3390/ijerph20032164_

Round 1

Reviewer 1 Report

Overall, the paper is interesting to read and it has provided information on the impacts of environmental regulations of countries along the Belt and Road Initiatives on carbon emissions in terms of technical innovation, industrial structure and energy intensity. However, rigorous analysis on the interrelationship between model variables and the impact outcomes can be significantly improved. In addition,  there are number of writing errors throughout the paper which must be corrected and long sentences should be shortened to clarify the sentences.

Please provide more clarity on the following issues:

- In abstract you have stated: Accordingly, we suggest that countries along the Belt and Road, to follow the road of sustainable and low-carbon development, should further enhance their focus on environment protection, improve their environmental awareness and take environmental regulation measures rationally to reduce carbon emissions : Elaborate what kind of activities are required in the conclusion section?

- There is lack of clear definitions of technical innovation and environmental regulation. Define them with examples. This is important to follow the paper. Is number of innovation patents a good variable for mapping the level of environmental regulations?

- Also, open up following phrases a bit more possibly with example: energy intensity and industrial structure.

Line 51: single industrial structure - what is it? Expand a bit more to make it clearer.

Line 55: Environmental regulation measures – What are these? Describe with examples.

Line 59 -62 –too lengthy sentence, consider revising?

Line 65-66: Environmental regulation is regarded as one of the important measures to tackle environment problems : environmental regulation be defined here with examples?

Line 82-83: …ARDL Model and found that environmental regulation and carbon emissions have negative correlations in both the short- and long- terms.    This is interesting to know!

Line 98: Green Paradox effects  - This can be described so that readers can understand it easily.

Line 120: Technical innovation - this should be elaborated.

Line 159-160: The effects on carbon emissions from environmental regulation cannot be simply summarized into promotion first and inhibition next …not clear?

Line 200: OLS Regression - Expand OLS

Line 419-421 – The unreasonable energy consumption structure featuring high emissions of countries along the Belt and Road contributes chiefly to the increased carbon emissions.? This sentence should be revised.

Line 524-525 -  Under different industrial structures : can you elaborate what are these different industrial structures?  

Line 684 - Environmental regulation measures – what kind of environmental regulation measures?

Reviewer 2 Report

This paper selects the relation between the carbon emission and environmental regulation as the study topic and has some theoretical and practical significance. The paper adds the institutional background of the Belt and Road into the research and has some marginal contributions to the literature. However, there are some concerns as follows.

(1) The paper should explain why to select the Belt and Road as the setting to explore the relation between the carbon emission and environmental regulation, and what are the differences?

(2) The paper uses the number of environmental technology invention patents as a measurement of environmental regulations. It may not be a perfect indicator. The authors should have some robust checks with alternative variables.

(3) The paper should study the difference among the countries along the Belt and Road.

(4)The citing of some literature in the paper is not correct.

Round 2

Reviewer 2 Report

My concerned problems are as follows.

(1) The references are cited in the manuscript only with the authors’ surname, instead of the full name.

(2) Authors should add more robust checks with alternative variables of ER.
